# Partial Resection of Spinous Process for the Elderly Patients with Thoraco-Lumbar Kyphosis: Technical Report

**DOI:** 10.3390/medicina57020087

**Published:** 2021-01-21

**Authors:** Hirohiko Inanami, Hiroki Iwai, So Kato, Yuichi Takano, Yohei Yuzawa, Kazuyoshi Yanagisawa, Takeshi Kaneko, Tomohide Segawa, Ko Matsudaira, Hiroyuki Oka, Masahito Oshina, Masayoshi Fukusima, Fumiko Saiki, Yasushi Oshima

**Affiliations:** 1Department of Orthopaedic Surgery, Inanami Spine and Joint Hospital, 3-17-5 Higashi-shinagawa Shinagawa-ku, Tokyo 140-0002, Japan; y-yuzawa@iwai.com (Y.Y.); kazuyoshi728@gmail.com (K.Y.); mogu184cm@gmail.com (T.K.); t.segawa@sand.ocn.ne.jp (T.S.); 2Department of Orthopaedic Surgery, Iwai Orthopaedic Medical Hospital, 8-17-2 Minamikoiwa Edogawa-ku, Tokyo 133-0056, Japan; h-iwai@iwai.com (H.I.); luigi.igiul1030@gmail.com (Y.T.); 3Department of Orthopaedic Surgery, The University of Tokyo, 7-3-1 Hongo, Bunkyo-ku, Tokyo 113-8655, Japan; sokato@g.ecc.u-tokyo.ac.jp (S.K.); oshinamasahito@gmail.com (M.O.); masamasayoshi21@hotmail.co.jp (M.F.); saikifum@yahoo.co.jp (F.S.); yoo-tky@umin.ac.jp (Y.O.); 422nd Century Medical & Research Center, Department of Medical Research and Management for Musculoskeletal Pain, The University of Tokyo, 7-3-1 Hongo, Bunkyo-ku, Tokyo 113-8655, Japan; kohart801@gmail.com (K.M.); taigis54@yahoo.co.jp (H.O.)

**Keywords:** lumbar kyphosis, spinous process, partial resection

## Abstract

*Background and objectives:* Global sagittal imbalance with lumbar hypo-lordosis can cause low back pain (LBP) during standing and/or walking. This condition has recently been well-known as one of the major causes of reduced health-related quality of life (HRQOL) in elderly populations. Decrease in disc space of anterior elements and an increase in the spinous process height of posterior elements may both contribute to the decrease in lordosis of the lumbar spine. To correct the sagittal imbalance, the mainstream option is still a highly invasive surgery, such as long-segment fusion with posterior wedge osteotomy. Therefore, we developed a treatment that is partial resection of several spinous processes of thoraco-lumbar spine (PRSP) and lumbar extension exercise to improve the flexibility of the spine as postoperative rehabilitation. *Materials and Methods:* Consecutively, seven patients with over 60 mm of sagittal vertical axis (SVA) underwent PRSP. The operation was performed with several small midline skin incisions under general anesthesia. After splitting the supraspinous ligaments, the cranial or caudal tip of the spinous process of several thoraco-lumbar spines was removed, and postoperative rehabilitation was followed to improve extension flexibility. *Results:* The average follow-up period was 13.0 months. The average blood loss and operation time were 11.4 mL and 47.4 min, respectively. The mean SVA improved from 119 to 93 mm but deteriorated in one case. The mean numerical rating scale of low back pain improved from 6.6 to 3.7 without any exacerbations. The mean Oswestry Disability Index score was improved from 32.4% to 19.1% in six cases, with one worsened case. *Conclusions:* We performed PRSP and lumbar extension exercise for the patients with LBP due to lumbar kyphosis. This minimally invasive treatment was considered to be effective in improving the symptoms of low back pain and HRQOL, especially of elderly patients with lumbar kyphosis.

## 1. Introduction

Global sagittal imbalance with lumbar hypo-lordosis has recently been well-known as one of the major causes of reduced health-related quality of life (HRQOL) in elderly populations [1,2]. This condition can cause a variety of problems, including low back pain (LBP) while standing and/or walking. This outcome is mainly because the back muscles become too exhausted to maintain an erect posture while the upper body spontaneously falls forward [2,3,4,5]. To correct the sagittal imbalance, highly invasive long-segment fusion surgery with posterior wedge osteotomy [6] is still the mainstream for these cases. Although the essence of these operations is the shortening of the posterior elements [7] and rationally correct, the procedures are highly invasive and frequently cause severe complications. Thus, development of minimally invasive procedures has been expected.

Decrease in disc space has been considered a well-known degenerative change that may decrease the lumbar lordosis, while an increase in the spinous process height of posterior elements has also been observed as one of the degenerative changes [8,9]. It has been reported that the cranio-caudal increase in size of the spinous processes as posterior elements may contribute to the decrease in lordosis of the lumbar spine [8]. That is, both shortening the height of the anterior element and the lengthening the height of the posterior element can lead to reducing the lordosis of the lumbar spine [10].

Therefore, we developed a treatment aiming (1) to shorten the length of the posterior element by partial resection of the spinous processes (PRSP) and (2) to improve the flexibility of the anterior element by lumbar extension rehabilitation.

## 2. Materials and Methods

This is a retrospective study of seven consecutive patients who underwent PRSP and lumbar extension exercise for thoraco-lumbar kyphosis between 1 March and 30 September 2019 at one single spine center.

### 2.1. Patients

The inclusion criteria were (1) chief complaint being LBP on standing or walking, (2) sagittal vertical axis (SVA) [11] > 60 mm on a standing radiograph, (3) more than some mobility of the anterior element and (4) inter spinous distance < 2 mm at three levels or more on mid-sagittal computed tomography image. The typical posture of the patients is shown in Figure 1. The exclusion criteria were as follows: (1) age < sixty years, (2) history of fusion operation, (3) scoliosis with lumbar curvature at a Cobb angle > 15°, (4) lumbar spinal stenosis on magnetic resonance imaging, (5) lumbar zygapophysial arthrosis [12], (6) sacro-iliac arthrosis [13] and (7) spondylolisthesis Meyerding grade ≥ II. These criteria were diagnosed radiologically or by physical findings or relevant block procedures if needed. The demographic data of the study population is shown in Table 1.

### 2.2. Operation Procedure

A longitudinal skin incision of 20 mm was made on every two spinous processes. The subcutaneous tissue was dissected. The supraspinous ligament was longitudinally separated. The periosteum was dissected to expose the spinous process and the multifidus muscle was preserved. The interspinous ligament was incised. A chisel was used to excise the cranial or caudal tip of the spinous process: 10 mm long, 10 mm wide and 20 mm deep. The ligament was re-sutured, and the skin was closed (Figure 2, Figure 3 and Figure 4).

### 2.3. Postoperative Rehabilitation

Patients were instructed to perform six sets of stretch training each day for ten seconds in the thoraco-lumbar extension position. This exercise consisted of training to push up the upper body with both upper limbs in the prone position. Additionally, a rolled towel was placed under the patients’ back on the supine position and the lumbar spine was extended and held for ten seconds. Each exercise was performed gradually, with care being taken not to cause severe pain. Compliance was monitored by physiotherapists checking the daily form filled out by patients themselves. The performance was tested at three weeks and every three months thereafter.

### 2.4. Measurements and Evaluation

A 64-slice computed tomography (CT) scanner (Discovery 750 HD/Revolution GSI; GE Healthcare, Tokyo, Japan) was used. The original thickness of the CT images was 0.625 mm, and the re-formatted slice thickness was 2.0 mm. All measurements were performed using imaging software (DICOM Image Work-Station XTREK F.E.S.T. A system; J-Mac System Inc., Sapporo, Japan).

The operation time, blood loss and complications were assessed. The radiographic and clinical findings were compared before surgery and postoperatively. The radiographic assessment consisted of SVA, lumbar lordosis (LL) and pelvic incidence (PI) at the standing position. The clinical evaluation was performed at minimum twelve months postoperatively, with patient-reported outcomes measures, including numerical rating scale (NRS) of LBP and leg pain as well as Oswestry Disability Index (ODI).

### 2.5. Ethical Standards

All procedures were performed in accordance with the ethical standards of the research committee of Iwai Medical Foundation and ethical approval of the committee was obtained (No. 20180926-1). Also, the study protocol was registered at the Japan Medical Association’s Knowledge Center on 27 February 2019 (JMA-llA00411). Informed consent was obtained by the disclaimer documents for the surgical procedure and rehabilitation methods handed over to the patient with explanations and signed. Specifically, we explained the advantages and disadvantages of this operation method, the mechanism of this disease group we supposed, possible complications and other treatment options. After that, we performed this operation on the patients who gave their consent. 

### 2.6. Statistical Analysis

We performed statistical analysis using SPSS 24.0 (SPSS Inc., Chicago, IL, USA). Wilcoxon Matched Pairs Test for NRS of LBP was performed pre- and post-operatively and ODI score was obtained. All reported *p*-values are two-tailed, with differences reported as significant when *p* < 0.05. An intra-class correlation coefficient (ICC) was calculated to explore consistency within and between examiners for measurements with the pre- and post-treatment images. To evaluate intra-rater reliability, we compared the measurements obtained by three examiners, and measured all the parameters of the seven subjects twice with more than a one-week interval. To evaluate inter-rater reliability, we compared the mean of two measurements for all parameters of the same seven subjects. An ICC value approaching 1.0 indicates less variability and better consistency, and a value over 0.8 is considered sufficiently reliable.

## 3. Results

### 3.1. Reliability of the Measurements in the Pre- and Post-Treatment Radiograph

The ICCs of intra-rater reliabilities of three raters for five measurements were from 0.900 to 0.999, and the ICCs of inter-rater reliabilities were from 0.932 to 0.996 (Table 2). Overall, the ICC data suggested excellent measurement consistency and reliability in both the pre- and post-treatment.

### 3.2. Operation Detail

Of the eight cases who underwent PRSP, seven cases were followed up for twelve months or more after we excluded one case whose daily activity became limited only due to being bedridden due to Parkinson’s syndrome during the follow-up period. The age at surgery was 75 to 87 years old, with the average being 79.9 years, and the average follow-up period was 13.1 months. Surgical time ranged from 34 to 67 min, with an average of 47 min. Several sites of Th11-S1 spinous processes were excised. Six sites were involved in four cases, five sites in one case and four sites in two cases. Intraoperative bleeding was 10 to 20 g, with an average of 11.4 g. There was no postoperative muscle weakness nor nerve palsy. There were no other serious complications, but one case experienced delayed wound healing, and the wound finally closed three weeks after the operation. Facet joint disease developed in two cases a few months after PRSP. The cauterization of posteromedial branch of existing nerve roots going to the corresponding facet joint [12] was performed in one case (case No. 3). The details of each operation are shown in Table 3.

### 3.3. Radiographical Outcome

SVA improved from an average of 119.2 to 93.4 mm (*p* < 0.05) (Table 4). Deterioration (+54.9 mm) was observed in one case, almost unchanged (−0.5 mm) in one case and the remaining five cases improved by 47.2 mm on average. LL in the standing position was also improved except for one case. CT and radiography of illustrative case are shown in Figure 5 and Figure 6.

### 3.4. Patient-Oriented Outcomes

NRS of LBP improved from an average of 6.6 to 3.7 (*p* < 0.05) (Table 5). Significant improvement of eight points and five points lower change were observed in two case (No.3, No.7), while other cases showed mild improvement ranging from one to two points without any worsening cases. Lower extremity pain did not change on average. ODI score improved from pre-treatment average of 32.4% to postoperative average of 19.1% (*p* < 0.05). Given that the minimum clinically important differences of ODI, NRS of LBP and leg pain in lumbar spine surgery patients have been reported as 12.8, 1.2 and 1.6 respectively [14], five cases out of seven showed clinically significant improvement in NRS of LBP, and four out of seven showed improvement in HRQOL by ODI.

## 4. Discussion

Many patients have undergone long-segment spinal fusion surgeries with osteotomy for the correction of sagittal imbalance [5]. These highly invasive operations may cause various complications and a wide range of rigid spines. The key concept is to shorten the posterior elements with or without elongation of anterior elements as spinal kyphosis is primarily caused by anterior column height collapse. However, in addition to anterior shortening, it has been advocated that enlargement of posterior elements such as spinous processes can be another source of kyphosis by blocking the back extension in some cases [10]. Thus, we developed minimally invasive procedures, that is, PRSP to shorten the posterior elements without fusion operation and lumbar extension exercise to increase the flexibility of anterior elements. The present case series with technical description is the first report of PRSP for spinal flexible kyphosis, to the best of our knowledge.

The invasiveness of the surgery was considerably minimum: SVA was markedly improved (>45 mm) in four out of seven cases (57%), with LL also being recovered in five out of seven cases (71%), which showed that our concept of posterior shortening was successful in the majority of the cases selected. As a result, LBP significantly improved in five and ODI in four out of seven cases.

As a cause of back pain during standing or walking in kyphosis patients, fatigue pain for maintaining erect position against the tendency to fall forward has been previously discussed [2,3,4,5]. Therefore, our primary goal was to decrease SVA, aiming for unloading the back muscle to improve the back pain. However, interestingly, in this case series, low back pain improved even in cases where SVA did not improve. There are several explanations for this result. The positive relationship between LBP and paraspinal muscle pressure has been reported [15]. Another report argued that one of the parameters that influence the ability to maintain center of mass within “cone of economy” [5] is flexibility of the spine [16]. In summary, the following three were considered to be the mechanisms of improvement of LBP by this treatment method: (1) improvement of SVA reducing the load of the lumbar extensor muscles, (2) decreased paraspinal muscle pressure with decreasing of the compartment by PRSP and (3) improved flexibility of the spinal column contributing to optimized standing balance.

Invasive surgery is often challenging for patients with global sagittal imbalance, to whom this treatment is indicated. However, the present technique is minimally invasive and has very few and minor complications, while having a significant effect on reducing LBP, and it may be one of the promising options for thoraco-lumbar kyphosis in the elderly.

The effect of PRSP is not so dramatic in this study. One of the reasons is the rigid anterior elements. Therefore, in order to know the actual therapeutic effect of PRSP, it should be considered to apply PRSP to younger patients who have soft anterior elements. For this purpose, it is necessary to investigate what kind of patients cause kyphosis of the thoracolumbar spine due to hypertrophy of the posterior element (spinous processes) alone. 

This study has some limitations. First, the follow-up period is relatively short, and the minimum follow-up period was 12 months in the present case series. Long-term outcomes should be confirmed with continuous observation. Second, the number of cases included was still limited. Thirdly, patients with mild lumbar kyphosis were not included. Symptoms related to kyphosis usually only develop after deformity becomes moderate or severe. It may be important to elucidate whether anterior shortening or posterior augmentation should precede to determine the appropriate decision-making and timing for the treatment of this disease at early or advanced stages. Further study is warranted. 

## 5. Conclusions

We performed PRSP and lumbar extension exercise for the patients with LBP during standing and/or walking due to lumbar kyphosis. This treatment method was minimally invasive and was considered to be effective in improving the symptoms of low back pain and HRQOL, especially of elderly patients. For the full evaluation of the effectiveness of this treatment, it is necessary to investigate a large number of cases and to study the longer-term outcome.

## Figures and Tables

**Figure 1 medicina-57-00087-f001:**
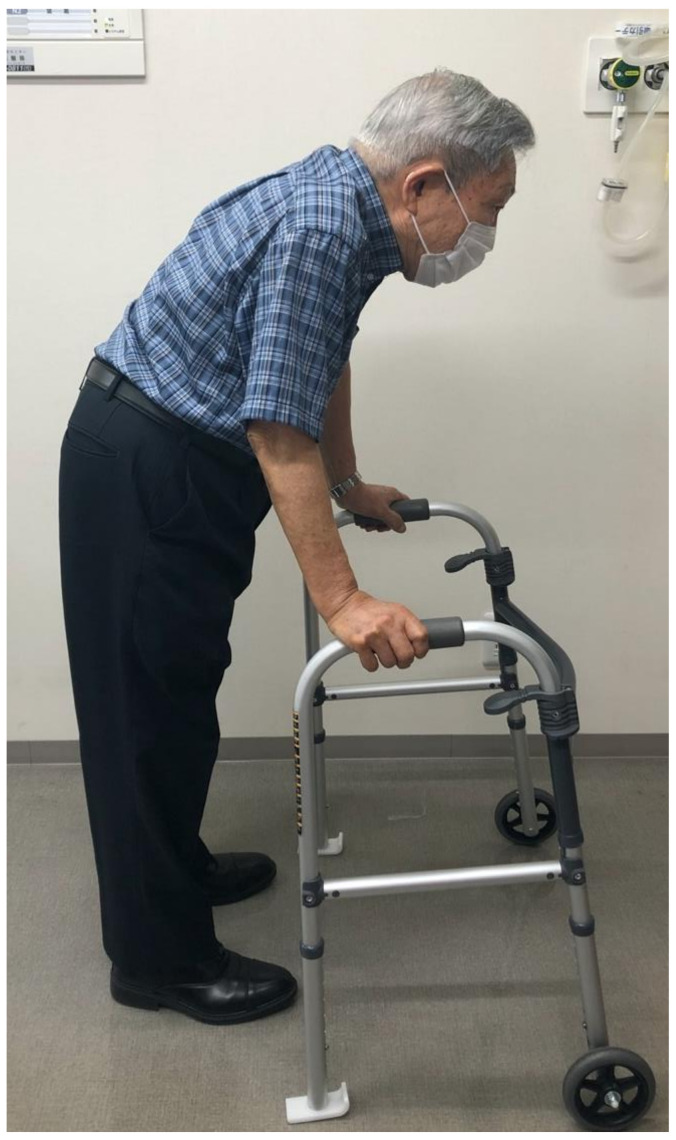
Typical gait of this group of patients. These patients often use walking aids.

**Figure 2 medicina-57-00087-f002:**
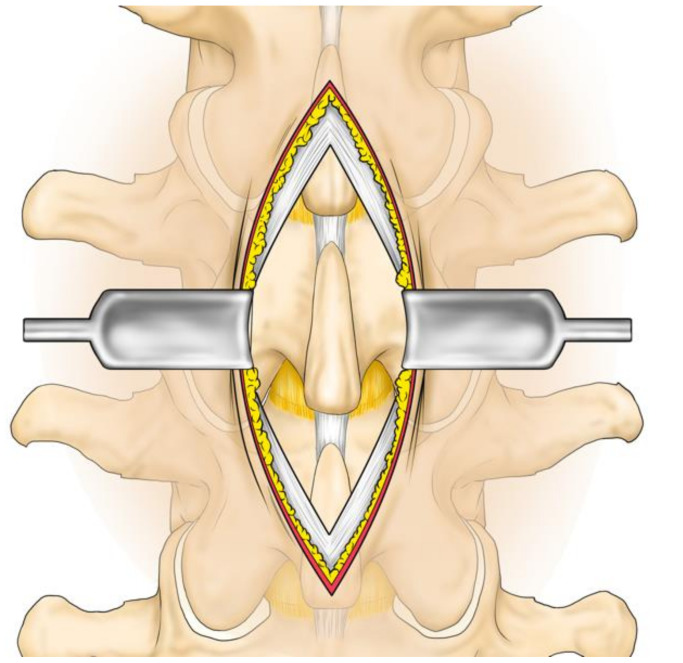
Operation procedure 1: exposure of spinous process. The skin and the supra-spinous ligament are incised, and the spinous process is exposed. In the actual surgery, a 2 cm skin incision is made, the skin is displaced and two spinous processes are partially excised.

**Figure 3 medicina-57-00087-f003:**
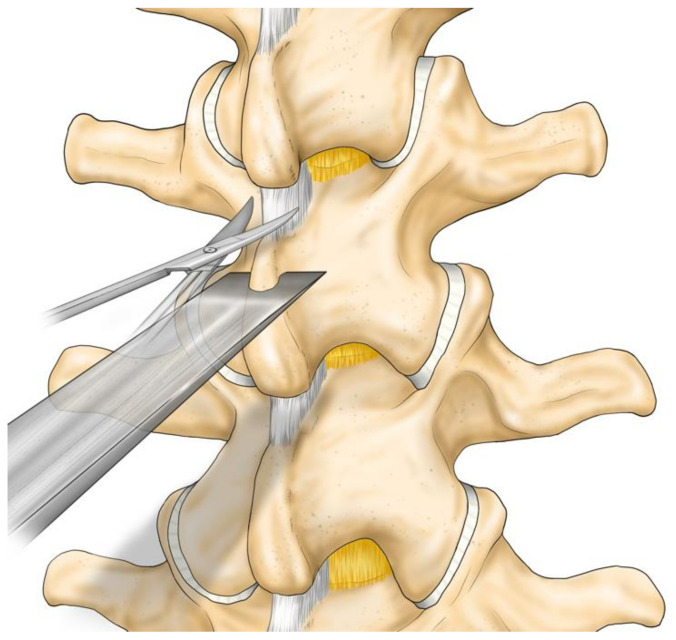
Operation procedure 2: excision of the cranial part of the spinous process. Incise the inter-spinous ligament. A chisel strikes the spinous process.

**Figure 4 medicina-57-00087-f004:**
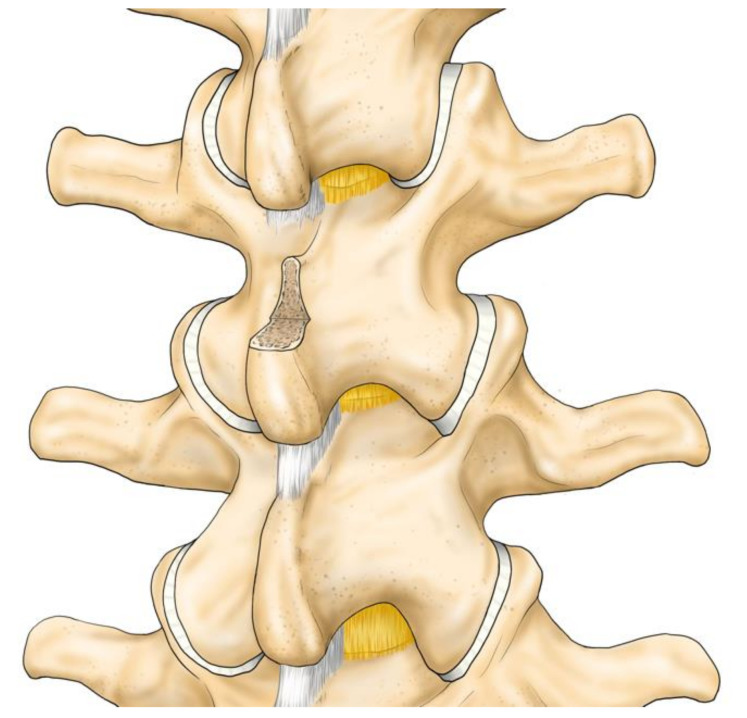
Operation procedure 3: The cranial portion of the spinous process is removed.

**Figure 5 medicina-57-00087-f005:**
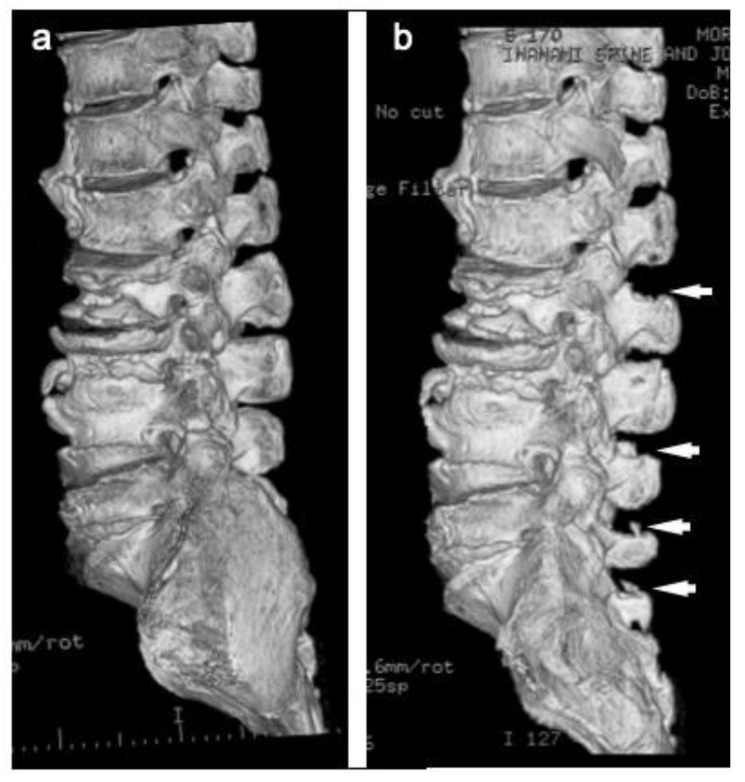
Computed tomography (CT) of illustrative case at pre- and post-operation (Case #4). (**a**) Pre-treatment image, (**b**) post-treatment image. The cranial portion of spinous process of L2, 4, 5 and S1 were resected (white arrows).

**Figure 6 medicina-57-00087-f006:**
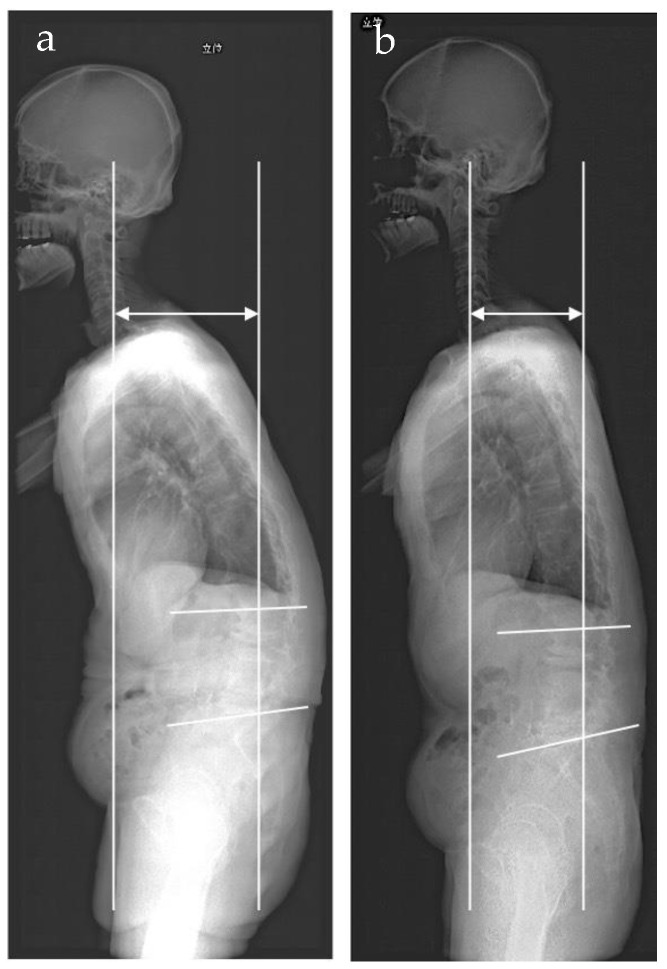
Sagittal vertical axis (SVA) and lumbar lordosis (LL) on radiographs of total spine on standing position (Case #4). (**a**) pre-treatment image, SVA: 154 mm, LL: 6.8°. (**b**) Post-treatment image, SVA: 108.1 mm, LL: 12.8°.

**Table 1 medicina-57-00087-t001:** Demographics.

Case No.	Sex	Age (Years)	Height (cm)	Body Weight (kg)	BMI
1	F	82	143.5	49.1	23.8
2	F	77	147.6	42.2	19.4
3	F	87	146.0	61.7	28.9
4	M	79	168.5	64.0	22.5
5	F	77	149.0	57.5	25.9
6	F	75	152.6	56.6	24.3
7	M	82	160.5	80.7	31.3
average		79.9	152.5	58.8	25.2

F: female, M: male, BMI: body mass index.

**Table 2 medicina-57-00087-t002:** Comparison of intra-class correlation coefficient (ICC) and 95% confidential interval (CI) for all the parameters between three raters.

	Intra-Rater Reliability						Inter-Rater Reliability	
	Rater 1		Rater 2		Rater 3			
	ICC	95%CI	ICC	95%CI	ICC	95%CI	ICC	95%CI
Pre-SVA	0.997	0.983/0.999	0.998	0.992/1.000	0.997	0.984/0.999	0.996	0.986/0.999
Post-SVA	0.997	0.984/0.999	0.999	0.994/1.000	0.998	0.992/1.000	0.995	0.981/0.996
Pre-LL	0.985	0.926/0.997	0.998	0.988/1.000	0.999	0.992/1.000	0.978	0.923/0.996
Post-LL	0.900	0.576/0.982	0.994	0.970/0.999	0.988	0.992/1.000	0.932	0.745/0.987
PI	0.922	0.654/0.986	0.974	0.873/0.995	0.967	0.843/0.994	0.955	0.813/0.992

SVA: sagittal vertical axis, LL: lumbar lordosis, PI: pelvic incidence.

**Table 3 medicina-57-00087-t003:** Operation details.

Case No.	Level of PRSP	No. of PRSP	Operation Time (mins)	Blood Loss (g)	Postoperative Complication	Postoperative Facet Joint Arthrosis	Follow-Up (months)
1	Th12, L1, 2, 3, 4, 5	6	51	10	-	-	15
2	L1, 2, 3, 4, 5	5	37	10	+ (delayed wound healing)	+	12
3	Th11, 12, L1, 3, 4, 5	6	67	10	-	+ (treated by radiofrequency ablation)	16
4	L2, 4, 5, S1	4	54	10	-	-	13
5	L2, 3, 4, 5	4	46	20	-	-	12
6	L1, 2, 3, 4,5, S1	6	34	10	-	-	12
7	L1, 2, 3, 4, 5, S1	6	43	10	-	-	12
Average		5	47	11.4	-		13

PRSP: partial resection of spinous process.

**Table 4 medicina-57-00087-t004:** Radiographic outcome.

Case No.	SVA Pre (mm)	SVA Post (mm)	SVA Change (mm)	LL Pre (Degree)	LL Post (Degree)	LL Change (Degree)	PI (Degree)
1	137.0	137.5	−0.5	1.9	7.2	5.3	57.5
2	88.8	27.3	61.6	−11.2	−3.2	8.0	35.8
3	117.0	55.5	61.5	2.5	6.1	3.5	35.5
4	154.0	108.1	45.9	6.8	12.8	5.9	41.2
5	63.4	118.2	−54.9	25.5	15.9	−9.6	54.0
6	109.4	91.3	18.0	1.7	2.2	0.5	48.1
7	165.1	115.8	49.3	−13.2	2.3	15.5	46.9
average	119.2	93.4	25.8	2.0	6.2	4.2	45.6

SVA: sagittal vertical axis, LL: lumbar lordosis, PI: pelvic incidence.

**Table 5 medicina-57-00087-t005:** Patient-oriented outcome.

Case No.	NRS of LBP Pre	NRS of LBP Post	LBP Change	NRS of Leg Pain Pre	NRS of Leg Pain Post	Leg Pain Change	ODI Pre (%)	ODI Post (%)	ODI Change (%)
1	7	5	−2	7	6	−1	49	27	−22
2	8	6	−2	3	4	+1	20	22	+2
3	8	3	−5	0	0	0	56	40	−16
4	5	4	−1	0	0	0	31	18	−13
5	2	0	−2	0	0	0	9	0	−9
6	8	8	0	5	6	+1	31	27	−4
7	8	0	−8	3	0	−3	31	0	−31
Average	6.6	3.7	−2.4	2	2	0	32.4	19.1	−13.3

NRS: numerical rating scale, ODI: Oswestry Disability Index (%), LBP: low back pain.

## Data Availability

The datasets generated and analyzed during the current study are available from the corresponding author upon reasonable request.

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
