# Peer review of "Partial Resection of Spinous Process for the Elderly Patients with Thoraco-Lumbar Kyphosis: Technical Report"

_medicina, 2021, doi:10.3390/medicina57020087_

Round 1

Reviewer 1 Report

An interesting and clear presentation of an innovative, minimally invasive method of treating axial pain of the lumbar spine in the course of sagittal balance disorders. A short but clear form of presentation. It's nice to read. The disadvantage is the numerically small material and the short observation period, but as I understand it, the main assumption was to present the technique and not to evaluate its effectiveness. The expected effectiveness results from intellectually attractive pathophysiological assumptions. It will require verification in the future and it should be underlined in conclusion.

What I'm missing is a reference to the symptoms and root deficits in these patients and their changes (appearance) during treatment, for example:  neurogenic claudication. It is clear that in some of these patients, postural disorders are a consequence of reflex decompression of the narrow spinal canal, and that simple deformation correction without decompressing the nerve structures may exacerbate the symptoms of stenosis. I wonder how you exclude e stenotic patients from this procedure.

Author Response

Thank you for nice instructive comments. One of the keys to the success of this treatment is the rigorous selection of patients with fatigued low back pain due to kyphosis. Partial resection of spinous process(PRSP) tends to enhance the lordosis of the lumbar spine, and PRSP exacerbates the degree of stenosis. Therefore, patients with lumbar spinal stenosis should never be included in this treatment. The diagnosis of lumbar spinal stenosis is made by imaging and clinical symptoms. The diagnostic imaging is that there is no stenosis in the central canal and lateral recess of the spinal canal on MRI, and that there is no stenosis inside or outside the intervertebral foramen. On the other hand, clinical symptoms include cauda equina and nerve root symptoms. That is, pain in the thighs and lower legs when standing or walking, numbness, and weakness. The target patients for this treatment are strictly selected cases without any stenosis.

We would like to add the following sentence to the Conclusions.

“For the fully evaluation of the effectiveness of this treatment, it is necessary to investigate a large number of cases and to study the longer-term outcome.”

We would like to add the following sentence to the Discussion, also.

“The effect of PRSP is not so dramatic in this study. The one of the reasons is the rigid anterior elements. Therefore, in order to know actual therapeutic effect of PRSP, it should be considered to apply PRSP to younger patients who have soft anterior elements. For this purpose, it is necessary to investigate what kind of patients cause kyphosis of the thoracolumbar spine due to hypertrophy of posterior element (spinous processes) alone.”

Reviewer 2 Report

Did authors quantify change in FSU motions pre and post surgery and at follow up?  It will be a good data.  How much resection of the spinous processes did authors undertake?  What was the basis for making this decision?  Did the amount resection vary with spinal levels?

Author Response

Thank you for nice instructive comments, but we did not quantify in FSU motions pre and post surgery. Therefore, we would like to investigate it in the next research.

The intended amount of resection was 10 mm long, 10 mm wide, and 20 mm deep (Fig. 3,4). We resected the spinous process partially as the following table.

Th11

Th12

L1

L2

L3

L4

L5

S1

Length

Average

4.3

8.7

8.7

10.2

9.1

10.6

9.4

6.4

Max

4.3

9.6

11.3

16.7

12.1

13.4

13.2

10.6

Min

4.3

7.7

6.1

4.8

6.1

7.4

3.2

0.5

Deepness

Average

17.4

7.9

15.0

15.1

19.0

19.6

18.7

13.2

Max

17.4

8.7

20.5

27.6

25.8

31.7

23.7

15.2

Min

17.4

7.1

7.0

5.7

7.7

12.5

10.9

9.7

The amount of excised bone was within the range that did not impede the movement of the corresponding FSU and the spinous process did not break. It may change slightly depending on the shape of the spinous process.

Round 2

Reviewer 2 Report

Authors have addressed this reviewer's concerns in a reasonable manner